# Computing Low-Entropy Couplings for Large-Support Distributions

**Samuel Sokota**[1]     **Dylan Sam**[1]     **Christian Schroeder de Witt**[2]     **Spencer Compton**[3]

**Jakob Foerster**[2]     **J. Zico Kolter**[1,4]

[1]Carnegie Mellon University
[2]University of Oxford
[3]Stanford University
[4]Bosch AI
ssokota@andrew.cmu.edu

## Abstract

Minimum-entropy coupling (MEC)—the process of finding a joint distribution with minimum entropy for given marginals—has applications in areas such as causality and steganography. However, existing algorithms are either computationally intractable for large-support distributions or limited to specific distribution types and sensitive to hyperparameter choices. This work addresses these limitations by unifying a prior family of iterative MEC (IMEC) approaches into a generalized partition-based formalism. From this framework, we derive a novel IMEC algorithm called ARIMEC, capable of handling arbitrary discrete distributions, and introduce a method to make IMEC robust to suboptimal hyperparameter settings. These innovations facilitate the application of IMEC to high-throughput steganography with language models, among other settings. Our codebase is available at https://github.com/ssokota/mec.

## 1 INTRODUCTION

Given two marginal distributions, a coupling is a bivariate joint distribution with the given marginals. In general, there may be many couplings for a particular pair of marginals. The problem of computing a coupling with the minimum amount of joint entropy among all feasible couplings is called minimum-entropy coupling (MEC) [Kovačević et al., 2015]. Further detailed in Compton et al. [2023], applications of MEC include causal inference [Kocaoglu et al., 2017, Compton et al., 2020, Javidian et al., 2021, Compton et al., 2022], communication [Sokota et al., 2022], steganography [Schroeder de Witt et al., 2023], random number generation [Li, 2021], multimodal learning [Liang et al., 2023], functional representations [Cicalese et al., 2019], and dimensionality reduction [Vidyasagar, 2012, Cicalese et al., 2016].

While MEC is NP-hard [Kovačević et al., 2015], recent works have provided approaches that achieve provable approximations of MECs [Kocaoglu et al., 2017, Cicalese et al., 2019, Rossi, 2019, Li, 2021, Compton, 2022, Compton et al., 2023, Shkel and Yadav, 2023] in log-linear time (i.e., $O(N \log N)$) in the cardinality of the support of the marginals. Unfortunately, the supports of many distributions of interest, such as those of generative AI models, are intractably large for these provable approximation algorithms.

To handle such cases, Sokota et al. [2022] introduced a class of heuristic algorithms for producing low-entropy couplings. These algorithms work by iteratively coupling components of random vectors using provable MEC approximation algorithms in such a way that guarantees the aggregate joint distribution is a coupling. In practice, both Sokota et al. [2022] and Schroeder de Witt et al. [2023] find that these iterative minimum-entropy coupling (IMEC) approaches produce low-entropy couplings for distributions with very large supports—binary images and trajectories of Atari games [Bellemare et al., 2013] in the work of Sokota et al. [2022] and binary strings and generative models (including GPT-2 [Radford et al., 2019], WaveRNN [Kalchbrenner et al., 2018], and Image Transformer [Parmar et al., 2018]) in the work of Schroeder de Witt et al. [2023]. Unfortunately, the applicability of the IMEC algorithms Sokota et al. [2022] introduced is limited to problems in which one distribution either has small support or is factorable. Moreover, these algorithms can be sensitive to hyperparameter choices, requiring careful tuning for optimal performance. *As a result, at the time of writing, there exist no techniques for producing low-entropy couplings of general large-support distributions*, let alone any that are also robust to hyperparameter settings.

In this work, we make multiple contributions regarding the IMEC line of research. First, we unify existing IMEC algorithms under a single formalism using *sets of partitions*, where each partition is over the sample space of one of the given marginals. IMEC couples distributions by iteratively performing (approximate) MECs between a conditional

distribution of one marginal and the posterior over the blocks of a partition associated with the other marginal. In particular, at each iteration, IMEC uses a partition whose associated posterior maximizes entropy.

Leveraging this formalism, we derive the first algorithm for computing low-entropy couplings for arbitrary large-support distributions, which we call autoregressive IMEC (ARIMEC). ARIMEC uses a set of partitions, which we call the prefix tree partition set, in which each partition corresponds to a node of the prefix tree of one of the sample spaces. These prefix trees can have large numbers of nodes (and thereby induce large numbers of partitions). Thus, to facilitate an efficient implementation, we introduce techniques to 1) lazily update the posterior over different blocks and 2) quickly search over partitions using pruning.

Finally, recognizing IMEC's general brittleness to partition set choice, we introduce a technique, called merging, to improve its robustness. At each iteration, this technique merges sample realizations into groups with identical posterior updates. Merging uses these groupings to perform additional MECs when entropy would otherwise be wasted due to suboptimal partition sets or other factors.

We empirically validate the utility of our innovations in two settings: Markov coding games [Sokota et al., 2022] and steganography [Cachin, 1998]. In Markov coding games, the objective is to encode messages into the trajectories of a Markov decision process while achieving a high expected return. In steganography, the goal is to embed sensitive information into innocuous content such that an adversary cannot detect the hidden information. Our results show that ARIMEC achieves substantially improved communication rates in both settings, illustrating its ability to use autoregressive prior information about realistic messages. Additionally, we demonstrate that merging significantly enhances IMEC's robustness to suboptimal partition set choices, thereby facilitating easier out-of-the-box application. Overall, our results suggest ARIMEC with merging as a practical approach to applications involving computing low-entropy couplings for large-support distributions, such as high-throughput steganography with language models.

## 2 BACKGROUND AND NOTATION

For our background, we formally introduce minimum-entropy coupling and discuss existing techniques for computing and (heuristically) approximating minimum-entropy couplings. Thereafter, we introduce notation for partitions of sets, which we will later use to unify existing methods in one general framework.

### 2.1 MINIMUM-ENTROPY COUPLING

We begin by formalizing the ideas of couplings and minimum-entropy couplings.

**Definition 2.1** (Coupling). *Let $\mu \colon \mathbb{X} \to [0, 1]$ be a probability distribution over a finite set $\mathbb{X}$ and let $\nu \colon \mathbb{Y} \to [0, 1]$ be a probability distribution over a finite set $\mathbb{Y}$. A coupling of $\mu$ and $\nu$ is a bivariate joint probability distribution $\gamma \colon \mathbb{X} \times \mathbb{Y} \to [0, 1]$ that marginalizes to $\mu$ and $\nu$. In other words, $\gamma$ satisfies*

$$\sum_{x' \in \mathbb{X}} \gamma(x', y) = \nu(y), \text{ for all } y \in \mathbb{Y}, \qquad (1)$$

$$\sum_{y' \in \mathbb{Y}} \gamma(x, y') = \mu(x), \text{ for all } x \in \mathbb{X}. \qquad (2)$$

*We use $\Gamma(\mu, \nu) = \{\gamma \mid \gamma \text{ satisfies (1) \& (2)}\}$ to denote the set of all couplings for $\mu$ and $\nu$.*

**Definition 2.2** (Joint Entropy). *Given a coupling $\gamma$, the joint entropy is defined as*

$$\mathcal{H}(\gamma) = -\mathbb{E}_{(X,Y) \sim \gamma} \log \gamma(X, Y).$$

Throughout the paper, we will use capital letters to denote random variables, as is done in the definition above.

**Definition 2.3** (Minimum-Entropy Coupling). *Given two marginal distributions $\mu, \nu$, a **minimum-entropy coupling** is a coupling $\gamma \in \Gamma(\mu, \nu)$ such that*

$$\mathcal{H}(\gamma) = \min\{\mathcal{H}(\gamma') \mid \gamma' \in \Gamma(\mu, \nu)\}.$$

### 2.2 COMPUTING AND APPROXIMATING MINIMUM-ENTROPY COUPLINGS

While computing an exact minimum-entropy coupling is NP-hard [Kovačević et al., 2015], there has been a series of recent works that construct $O(N \log N)$ approximation algorithms, where $N$ is the size of the sample space. Cicalese et al. [2019] introduce an approximation algorithm that they show guarantees a coupling within 1 bit of minimum entropy. Rossi [2019] show that Kocaoglu et al. [2017]'s greedy approach guarantees a coupling within 1 bit of minimum entropy. Li [2021] introduce a third approach for which he also proved a 1 bit approximation guarantee. Most recently, Compton et al. [2023] show an improved guarantee for Kocaoglu et al. [2017]'s greedy approach of about 0.53 bits, while also showing that Cicalese et al. [2019] and Li [2021]'s algorithms cannot match this guarantee. Compton et al. [2023] also give approaches that guarantee exact MECs, though they require exponential time.

**Algorithm 1** Tabular IMEC: $Y \mid X = x$
***

**procedure** TIMEC$(\mu, \nu, x)$
    $\gamma(X) \leftarrow \mu(X)$
    **for** $j = 1, \ldots, m$ **do**
        $\gamma(X, Y_j \mid Y_{1:j-1}) \leftarrow \text{MEC}(\gamma(X \mid Y_{1:j-1}),$
                                    $\nu(Y_j \mid Y_{1:j-1}))$
        $Y_j \sim \gamma(Y_j \mid x, Y_{1:j-1})$
    **end for**
    return $Y$
**end procedure**

---

**Algorithm 2** Factored IMEC: $Y \mid X = x$
***

**procedure** FIMEC$(\mu, \nu, x)$
    $\gamma(X) \leftarrow \mu(X)$
    **for** $j = 1, \ldots, m$ **do**
        $i^* \leftarrow \operatorname{argmax}_i \mathcal{H}(\gamma(X_i \mid Y_{1:j-1}))$
        $\gamma(X_{i^*}, Y_j \mid Y_{1:j-1}) \leftarrow \text{MEC}(\gamma(X_{i^*} \mid Y_{1:j-1}),$
$\nu(Y_j \mid Y_{1:j-1}))$
        $\gamma(X, Y_j \mid Y_{1:j-1}) \leftarrow \gamma(X_{i^*}, Y_j \mid Y_{1:j-1}) \cdot$
                        $\prod_{i \neq i^*} \gamma(X_i \mid Y_{1:j-1})$
        $Y_j \sim \gamma(Y_j \mid x, Y_{1:j-1})$
    **end for**
    return $Y$
**end procedure**

---

## 2.3 ITERATIVE MINIMUM-ENTROPY COUPLING WITH A TABULAR POSTERIOR

In some settings, it is desirable to (non-provably) approximate minimum-entropy couplings where one random variable is a vector that ranging over such a large number of possible outcomes that the approaches described in Section 2.2 are inapplicable. Sokota et al. [2022] propose an iterative approach to such settings that assumes that this random vector is autoregressively specified. In this work, we refer to Sokota et al. [2022]'s algorithm as tabular IMEC (TIMEC). TIMEC guarantees that the resulting joint distribution is a coupling, supports conditional sampling and likelihood queries for both $X \mid Y$ and $Y \mid X$, where $Y$ is the random vector, and heuristically achieves low entropy. It can either be defined using the conditional generative process for sampling $Y \mid X$ or the conditional generative process for sampling $X \mid Y$, as both induce the same joint distribution. We focus on the process for generating $Y \mid X$, which is formalized in Algorithm 1, in the main body but include the process for generating $X \mid Y$ in Algorithm 5 in Appendix A. Algorithm 1 works iteratively in two steps:

1. First, it performs an (approximate) MEC between the posterior over $X$ given $Y_{1:j-1}$ (inductively defined via Bayes' Theorem) and the conditional distribution $\nu(Y_j \mid Y_{1:j-1})$.[1] The joint posterior over $X$ and $Y_j$ given $Y_{1:j-1}$ is assigned to the output of this coupling.

2. Second, it samples $Y_j$ from the posterior over $Y_j$ given both $X = x$ and $Y_{1:j-1}$ (also inductively defined via Bayes' Theorem).

## 2.4 ITERATIVE MINIMUM-ENTROPY COUPLING WITH A FACTORED POSTERIOR

Unfortunately, requiring approximate MECs over distributions of size $|\mathbb{X}|$ makes TIMEC inapplicable to many settings, such as steganography with large message sizes [Schroeder de Witt et al., 2023]. To ameliorate this issue, Sokota et al. [2022] also proposed a second approach, which

we refer to as factored IMEC (FIMEC)[2], in which $X$ is also assumed to be a random vector. Furthermore, crucially, it is assumed to be factorable.

**Assumption 2.4** (Factorability). *$X = (X_1, \ldots, X_n)$ is a random vector with $\mu(x) = \prod_i \mu(x_i)$ for all $x \in \mathbb{X}$.*

As with TIMEC, FIMEC guarantees that the resulting distribution is a coupling, supports likelihood queries to both conditionals and the joint distribution, and heuristically achieves low entropy. It can similarly be defined in terms of either conditional generative process ($X \mid Y$ or $Y \mid X$). We again focus on the $Y \mid X$ case (Algorithm 2), and defer the $X \mid Y$ case to Appendix A. The basic structure of Algorithm 2 is analogous to that of Algorithm 1. However, rather than performing MECs using $\gamma(X \mid Y_{1:j-1})$, FIMEC uses $\gamma(X_{i^*} \mid Y_{1:j-1})$, where $X_{i^*}$ is a component of $X$ with maximum posterior entropy. The other components $X_i$ for $i \neq i^*$ are left independent of $Y_j \mid Y_{1:j-1}$.

## 2.5 PARTITIONS OF SETS

As discussed in the introduction, we will show the IMEC algorithms discussed in the previous two sections can be unified into a single algorithm using partitions over $\mathbb{X}$. We use the following definitions and notation for partitions.

**Definition 2.5** (Partition). *A partition $\mathcal{P}$ of a set $\mathbb{X}$ is a collection of blocks $\{\mathbb{B}_1, \ldots, \mathbb{B}_\ell\}$ where:*

1. *Each block is a subset of $\mathbb{X}$.*
2. *Each pair of distinct blocks has an empty intersection.*
3. *The union of blocks is equal to $\mathbb{X}$.*

**Definition 2.6** (Block Function). *For a partition $\mathcal{P}$ of a set $\mathbb{X}$, the block function $\mathcal{B}_\mathcal{P} \colon \mathbb{X} \to \mathcal{P}$ maps $x$ to the block of the partition of which it is an element. When $X$ is a random variable, we use $B_\mathcal{P} = \mathcal{B}_\mathcal{P}(X)$ to denote the block of $\mathcal{P}$, as a random variable, to which $X$ belongs.*

---

[1]Note that we use upper-bound-inclusive indexing, so $Y_{1:0} = ()$, $Y_{1:1} = (Y_1)$, $Y_{1:2} = (Y_1, Y_2)$, etc.

[2]Schroeder de Witt et al. [2023] use the name iMEC for this approach.

Note that the probability distribution of $B_{\mathcal{P}}$ is defined by

$$\mu(B_{\mathcal{P}} = \mathbb{B}) = \mu(X \in \mathbb{B}) = \sum_{x \in \mathbb{B}} \mu(X = x).$$

# 3 A UNIFICATION OF ITERATIVE MINIMUM-ENTROPY COUPLING

We are now ready to describe our unification of existing IMEC algorithms. In this unification, different instances of IMEC are specified using different sets of partitions

$$\mathfrak{P} \subset \{\mathcal{P} \mid \mathcal{P} \text{ is a partition of } \mathbb{X}\}.$$

Instances of this unified perspective guarantee that the resulting distribution is a coupling, support conditional and likelihood queries for both $X \mid Y$ and $Y \mid X$, and heuristically produce low entropy. We define this unified perspective to IMEC using the conditional generative process given in Algorithm 3, which samples from $Y|X$. (Equivalently, it is defined by the generative process given in Algorithm 7 in Appendix A, which samples from $X|Y$). Algorithm 3 works iteratively in three steps:

1. First, it computes a partition $\mathcal{P} \in \mathfrak{P}$ inducing a maximum-entropy posterior. The entropy induced by a partition $\mathcal{P}$ at iteration $j$ is defined in terms of the probabilities over the blocks of the partition under $\gamma$, given $Y_{1:j-1}$. That is,

$$\mathcal{H}(\gamma(B_{\mathcal{P}} \mid Y_{1:j-1}))$$
$$= -\sum_{\mathbb{B} \in \mathcal{P}} \gamma(X \in \mathbb{B} \mid Y_{1:j-1}) \log \gamma(X \in \mathbb{B} \mid Y_{1:j-1}).$$

   The intuition behind selecting the maximum-entropy partition is that it heuristically offers the opportunity to reduce the joint entropy by the largest amount.[3]

2. Second, it performs an (approximate) MEC between the posterior over the blocks of the chosen partition $\mathcal{P}$ and the conditional distribution $\nu(Y_j \mid Y_{1:j-1})$. The joint posterior over the block $B_{\mathcal{P}}$ and $Y_j$ given $Y_{1:j-1}$ is assigned to the output of this coupling.

3. Third, it samples $Y_j$ from the posterior over $Y_j$ given both the block $\mathcal{B}_{\mathcal{P}}(x)$ and $Y_{1:j-1}$.

---

[3]A justification is as follows. Recall that

$$\max(\mathcal{H}(C), \mathcal{H}(D)) \leq \mathcal{H}(C, D) \leq \mathcal{H}(C) + \mathcal{H}(D),$$

where $\mathcal{H}(C, D)$ achieves its upper bound when $C$ and $D$ are independent. Thus, the maximum reduction in joint entropy achievable by performing a coupling is upper bounded by

$$\mathcal{H}(C) + \mathcal{H}(D) - \max(\mathcal{H}(C), \mathcal{H}(D)) = \min(\mathcal{H}(C), \mathcal{H}(D)).$$

Therefore, maximizing $\mathcal{H}(C)$ maximizes an upper bound on the joint entropy reduction.

---

**Algorithm 3** IMEC (Generic Form): $Y \mid X = x$

> **procedure** IMEC($\mu, \nu, x, \mathfrak{P}$)
>   $\gamma(X) \leftarrow \mu(X)$
>   **for** $j = 1, \ldots, m$ **do**
>     $\mathcal{P} \leftarrow \arg\max_{\mathcal{P} \in \mathfrak{P}} \mathcal{H}(\gamma(B_{\mathcal{P}} \mid Y_{1:j-1}))$
>     $\gamma(B_{\mathcal{P}}, Y_j \mid Y_{1:j-1}) \leftarrow \text{MEC}(\gamma(B_{\mathcal{P}} \mid Y_{1:j-1}),$
>                             $\nu(Y_j \mid Y_{1:j-1}))$
>     $Y_j \sim \gamma(Y_j \mid \mathcal{B}_{\mathcal{P}}(x), Y_{1:j-1})$
>   **end for**
>   return $Y$
> **end procedure**

## 3.1 THEORY

The general form of IMEC possesses the following two properties, which reduce to the results of Sokota et al. [2022] as a special case.

**Proposition 3.1** (Coupling). *IMEC induces a coupling of $\mu$ and $\nu$.*

**Proposition 3.2** (Greediness). *If the partition of singletons is in $\mathfrak{P}$, IMEC approximately minimizes $\mathcal{H}(X, Y_{1:j})$ subject to $\mu, \nu, \gamma(X, Y_{1:j-1})$ on the $j$th iteration, for each $j$.*

Proofs for these statements are provided in Appendix B.2 and Appendix B.3, respectively.

The general form of IMEC also possesses the following runtime guarantee, which says that IMEC can be implemented efficiently whenever maximum-entropy posterior partition computation is efficient.

**Proposition 3.3** (IMEC Runtime). *Given a polynomial-time function for computing the maximum-entropy posterior partition, IMEC can be implemented in polynomial time in $\max_j |\mathbb{Y}_j|, \max_i |\mathbb{X}_i|, m, n$.*

We prove Proposition 3.3 in Appendix B.1.

## 3.2 SPECIAL CASES

**Tabular Posterior** To implement TIMEC using Algorithm 3, we can select the partition set $\mathfrak{P}$ to be the set of all partitions of $\mathbb{X}$. As per Lemma B.3, which is stated and derived in Appendix B.4, the partition of singletons (or a partition that is equivalent up to measure zero) will always be selected, as it achieves maximum entropy. Coupling with the partition of singletons is equivalent to coupling over the whole set, which is exactly what TIMEC does.

Using Proposition 3.3, we derive the following runtime guarantee for TIMEC in Appendix B.1.

**Corollary 3.1** (TIMEC Runtime). *TIMEC can be implemented in polynomial time in $\max_j |\mathbb{Y}_j|, |\mathbb{X}|, m$.*

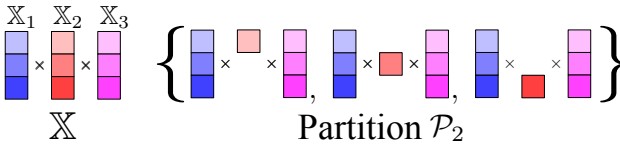

Figure 1: (Left) A set $\mathbb{X}$ of sequences of length 3; (right) a partition $\mathcal{P}_2$ used by FIMEC.

Note that TIMEC's polynomial time guarantee is in contrast to a direct application of an approximate MEC algorithm, which would require exponential time as a function of the same quantities.

**Factored Posterior**   To implement FIMEC using Algorithm 3, we can select the partition set as $\mathfrak{P} = \{\mathcal{P}_1, \ldots, \mathcal{P}_n\}$, where for each $i$,

$$\mathcal{P}_i = \{\mathbb{X}_1 \times \cdots \times \mathbb{X}_{i-1} \times \{x_i\} \times \mathbb{X}_{i+1} \times \cdots \times \mathbb{X}_n \mid x_i \in \mathbb{X}_i\}$$

and where $\mathbb{X}_i$ denotes the sample space for $X_i$. An example is shown in Figure 1. From this perspective, selecting $\mathcal{P}_i$ on a particular iteration is equivalent to selecting $X_i$ as the component with which to couple.

Using Proposition 3.3, we derive the following runtime guarantee for FIMEC in Appendix B.1.

**Corollary 3.2** (FIMEC Runtime).  *Let Assumption 2.4 hold. Then FIMEC can be implemented in polynomial time in* $\max_j |\mathbb{Y}_j|, \max_i |\mathbb{X}_i|, m, n$.

Note that FIMEC's polynomial-time guarantee is in contrast to a direct application of both an approximate MEC algorithm and TIMEC, which would each require exponential time in the same quantities.

# 4   ITERATIVE MINIMUM-ENTROPY COUPLING WITH AN AUTOREGRESSIVE POSTERIOR

Building on our unified framework, we now derive a new IMEC algorithm, which we call autoregressive IMEC (ARIMEC). ARIMEC improves upon the applicability of FIMEC by eliminating the factorability assumption. We present ARIMEC in two parts. First, we introduce the *prefix tree partition set*, which allows us to formally define ARIMEC using Algorithm 3. Second, we detail insights to make a practical implementation of ARIMEC.

## 4.1   MATHEMATICAL FORMALIZATION

In the framing of Algorithm 3, the defining characteristic of IMEC algorithms is their partition sets. Therefore, to develop an IMEC algorithm of maximal applicability, it is essential to choose a partition set compatible with a universal model of distributions (i.e., one capable of representing any distribution). The autoregressive model, which decomposes a distribution over vectors into component-wise conditional distributions via the chain rule of probability, is one such universal model. This section formalizes ARIMEC using a partition set specifically tailored to align with the tree-like output structure inherent in autoregressive models.

In order to define this partition set, which we call the prefix tree partition set, we first define prefixes.

**Definition 4.1** (Prefix/Extension).  *We write $v \sqsubset v'$ to mean that $v$ is a prefix of $v'$ in the substring sense and, equivalently, that $v'$ is an extension of $v$ in the substring sense.*

**Definition 4.2** (Immediate Prefix/Extension).  *We say $v$ is the immediate prefix of $v'$ and, equivalently, that $v'$ is the immediate extension of $v$, if $v \sqsubset v'$ and $v'$ is one character longer than $v$.*

Next, we define the *prefix tree* of a set of vectors. The prefix tree is a directed graph over prefixes of vectors with edges pointing to immediate extensions, as stated below. (Note that our usage of the term is graph theoretic and does not pertain to the trie data structure.)

**Definition 4.3** (Prefix Tree).  *The prefix tree for a set of vectors $\mathbb{X}$ is a directed graph $(\mathbb{V}, \mathbb{E})$, where the vertex set*

$$\mathbb{V} = \{v \sqsubset x \mid x \in \mathbb{X}\}$$

*is the set of prefixes of elements of $\mathbb{X}$ and the set of edges*

$$\mathbb{E} = \{(v, c) \mid v, c \in \mathbb{V}, c \text{ is an immediate extension of } v\}$$

*is the set of pairs of vertices and their immediate extensions. For distinct vertices $v, u$, we use the notation $\mathbb{V}_{v \to u}$ to mean the subset of $\mathbb{V} \setminus \{v\}$ touchable by paths that start from $v$ and contain $u$.*

We can view each vertex $v$ in the prefix tree as partitioning $\mathbb{X}$ in a manner that aligns with the sampling paths of autoregressive models. This perspective naturally induces what we call the prefix tree partition set, where each partition corresponds to a prefix upon which an autoregressive model could be conditioned.

**Definition 4.4** (Prefix Tree Partition Set).  *Let $(\mathbb{V}, \mathbb{E})$ be the prefix tree for $\mathbb{X}$. Then the **prefix tree partition set** is defined as $\mathfrak{P} = \{\mathcal{P}_v \mid v \in \mathbb{V}\}$, where*

$$\mathcal{P}_v = \{\mathbb{B}_{c \sqsubset} \mid (v, c) \in \mathbb{E}\} \cup \{\mathbb{B}_{v \not\sqsubset}\} \cup \{\mathbb{B}_{v=}\},$$

*and where*

- $\mathbb{B}_{c \sqsubset} = \{x \in \mathbb{X} \mid c \sqsubset x\}$ *denotes the subset of $\mathbb{X}$ that is an extension of the child $c$;*
- $\mathbb{B}_{v \not\sqsubset} = \{x \in \mathbb{X} \mid v \not\sqsubset x\}$ *denotes the subset of $\mathbb{X}$ that does not extend $v$;*

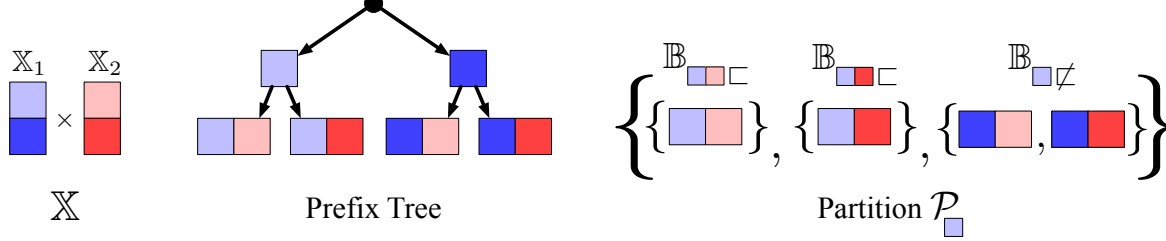

Figure 2: (Left) A set $\mathbb{X}$ of sequences of length 2; (middle) the prefix tree for $\mathbb{X}$; (right) the partition induced by the left-most depth one node of the prefix tree.

- $\mathbb{B}_{v=} = \{x \in \mathbb{X} \mid v = x\}$ *denotes the (either singleton or empty) subset of $\mathbb{X}$ equal to $v$.*

*For distinct vertices $v, u$, we use the notation $\mathbb{B}_{v \to u}$ to mean the subset of $\mathcal{P}_v \setminus \{\mathbb{B}_{v=}\}$ touchable by paths that start from $v$ and contain $u$.*

A visualization of the prefix tree and one partition that it induces is shown in Figure 2.

Having defined the prefix tree partition set, ARIMEC's formalization is immediate.

**Definition 4.5** (ARIMEC). *ARIMEC is the instance of Algorithm 3 in which the set of partitions $\mathfrak{P}$ is selected to be the prefix tree partition set.*

ARIMEC can be thought of as, at each iteration, operating at a working prefix $v$ whose associated partition $\mathcal{P}_v$ maximizes posterior entropy. As information is gained, the likelihood of one the blocks (such as some $\mathbb{B}_{c \sqsubset}$) will become large. As a result, the entropy associated with the working prefix's partition $\mathcal{P}_v$ will become small, causing the (entropy-maximizing) working prefix to change—often to a child of the existing working prefix. Over iterations, the working prefix will tend to traverse downward in the prefix tree toward the true value of $X$. However, it is also possible for it to move upward if the probability of $\mathbb{B}_{\not\sqsubset v}$ (for working prefix $v$) becomes large. This backtracking mechanism allows ARIMEC to recover from cases in which the working prefix deviates from prefixes of $X$.

We provide visual intuition for ARIMEC in Figure 3, showing example iterations for marginals of length two and its corresponding path down the tree in Figure 4.

## 4.2 EFFICIENT IMPLEMENTATION

While Definition 4.5 formalizes ARIMEC at a mathematical level, constructing a practical implementation is challenging due to the exponentially large number of nodes in the prefix tree, which makes naive maximum-entropy posterior partition computations intractable. To address this challenge, we propose a procedure that seeks to prove a maximum-entropy partition by searching over as few partitions as possible,

while lazily and (provably) efficiently computing the posterior (and posterior entropy) of each partition that it does search. This procedure has two components. The first is a polynomial time algorithm for lazily computing posteriors (and posterior entropies) for particular partitions. The second is a search procedure for finding a maximum-entropy partition that prunes partitions that are provably not maximum entropy. In practice, we observe that the procedure is highly efficient, often only requiring the evaluation of one or two nodes to prove a maximum-entropy partition, though we do not formally prove its runtime complexity; see Appendix D.1 for further details.

**Posterior Updates** The core idea behind our approach to posterior updates is that, given an updated posterior for the partition associated to node $v$, we can immediately derive the posterior of the partition for any adjacent node $u$. The posterior over $\mathcal{P}_u$ is dictated by two rules. First, that

$$\gamma(\mathbb{B}_{u \to v} \mid Y_{1:j}) = 1 - \gamma(\mathbb{B}_{v \to u} \mid Y_{1:j}),$$

by the complement law of probability. Second, that

$$\gamma(\mathbb{B} \mid Y_{1:j}) \propto \gamma(\mathbb{B} \mid Y_{1:j-1})$$

for $\mathbb{B} \in \mathcal{P}_u \setminus \mathbb{B}_{u \to v}$, as direct evidence about $B_{\mathcal{P}_v}$ does not differentiate between the elements of $\mathcal{P}_u \setminus \mathbb{B}_{u \to v}$. These ideas are formalized in Lemma B.4.

We can compute the posterior for any partition $\mathcal{P}_u$ in polynomial time by iteratively applying Lemma B.4 to the partitions along the undirected path from $v$ to $u$, as is stated below and proven in Appendix B.5.

**Proposition 4.1** (Posterior Updates). *Assume that the posterior over a partition is updated if and only if its corresponding node is touched and that nodes are touched by traversing edges of the tree (i.e., without jumps). Let $\mathcal{P}_v$ be a partition whose posterior was updated on iteration $j$. If $v, u$ are neighbors and $u$ was last visited on iteration $j' \le j$, then the iteration $j$ posterior for any partition $\mathcal{P}_u$ can be computed in polynomial time in $\max_i |\mathbb{X}_i|, n$.*

**Maximum-Entropy Posterior Partition Search** The core idea behind our search procedure is to prune nodes

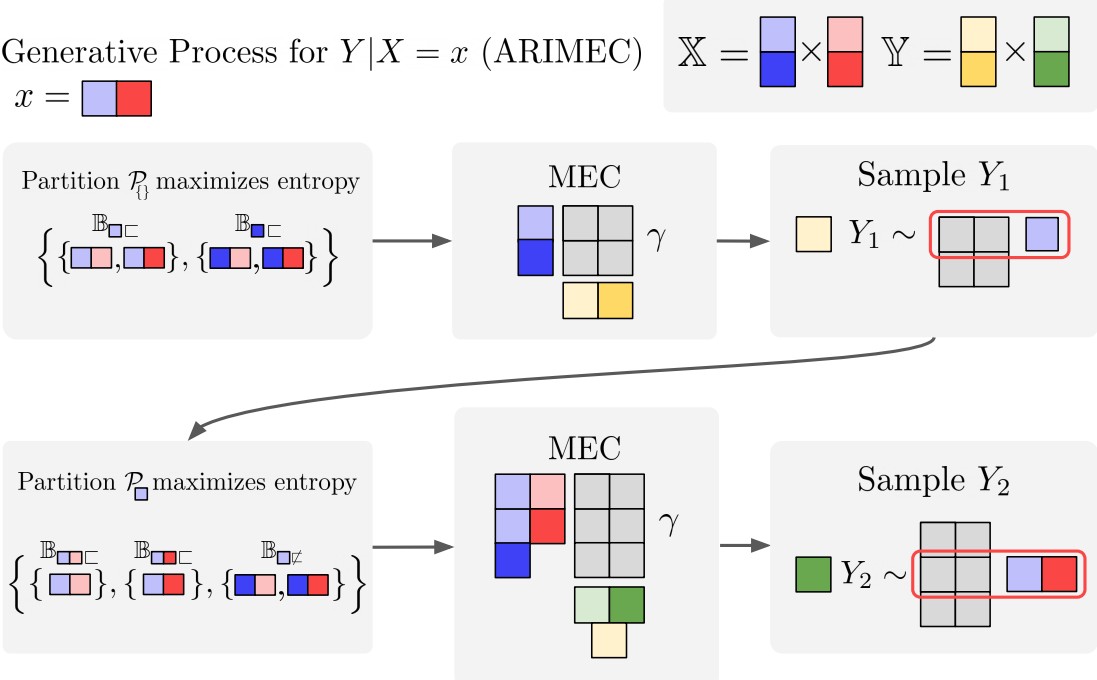

Figure 3: Visualization of two iterations of ARIMEC.

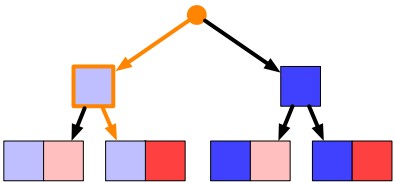

Figure 4: Path shown down the prefix tree corresponding to the procedure in Figure 3.

of the prefix tree whose partitions provably cannot be maximal entropy. To prune nodes, we make use of an upper bound on entropy. This upper bound—stated formally in Lemma B.5—shows, roughly speaking, that the entropy of any distribution with one sufficiently probable element cannot exceed the entropy of the distribution that would divide the remaining mass uniformly.

We can apply this upper bound on large numbers of nodes simultaneously: If a prefix $v$ is unlikely, then $\gamma(\mathbb{B}_{u \to v} \mid Y_{1:j})$ will be large for every $u$ in the subtree rooted at $v$. On the other hand, if a prefix $v$ is likely, then $\gamma(\mathbb{B}_{u \to v} \mid Y_{1:j})$ will be large for every $u$ in complement of the subtree rooted at $v$. We prove this result, stated in Proposition 4.2 below, in Appendix B.6.

**Proposition 4.2** (Maximum-Entropy Partition). *Let*

$$\kappa \geq \max_{\mathcal{P} \in \mathfrak{P}} |\{\mathbb{B} \in \mathcal{P} \mid \mu(\mathbb{B}) > 0\}|$$

*be an upper bound on the number of blocks with positive probability. Define*

$$\mathcal{U}: q \mapsto -q \log q - (1-q) \log \frac{1-q}{\kappa - 1}.$$

*For any neighbor $u$ of $v$, if*

$$\gamma(\mathbb{B}_{v \to u} \mid y_{1:j}) < 1 - 1/\kappa,$$

*then, for all $u' \in \mathbb{V}_{v \to u}$,*

$$\mathcal{H}(\gamma(B_{\mathcal{P}_{u'}} \mid y_{1:j})) \leq \mathcal{U}(\gamma(\mathbb{B}_{u \to v} \mid y_{1:j})).$$

Using Proposition 4.2, we can prove a maximum-entropy partition by searching only over the nodes for which we cannot prove an upper bound that is smaller than a previously observed entropy, as is described in Algorithm 4.

## 5 MITIGATING ENTROPY WASTE VIA MERGING

One suboptimality of ARIMEC, and more generally of all IMEC algorithms, results from the fact that, on a particular iteration, it may be the case that

$$\mathcal{H}(\nu(Y_j \mid Y_{1:j-1})) - \mathcal{H}(\gamma(B_{\mathcal{P}} \mid Y_{1:j-1})) > 0.$$

**Algorithm 4** Maximum-Entropy Partition Search

**procedure** MAXENTPARTITION($v, \gamma(\cdot \mid Y_{1:j})$)
    queue $\leftarrow [\mathcal{P}_v]$
    **while** queue non-empty **do**
        $\mathcal{P}_u \leftarrow$ queue.pop()
        **if** $\mathcal{H}(\gamma(B_{\mathcal{P}_u} \mid Y_{1:j}))$ is max ent so far **then**
            max ent partition $\leftarrow \mathcal{P}_u$
        **end if**
        **for** each node $u'$ adjacent to $u$ **do**
            $q \leftarrow \gamma(\mathbb{B}_{u \to u'} \mid Y_{1:j})$
            **if** $q > 1 - \frac{1}{\kappa}$ or $\mathcal{U}(q) >$ max ent so far **then**
                queue.append($\mathcal{P}_{u'}$)
            **end if**
        **end for**
    **end while**
    return max ent partition
**end procedure**

In such a case, IMEC necessarily wastes at least $\mathcal{H}(\nu(Y_j \mid Y_{1:j-1})) - \mathcal{H}(\gamma(B_{\mathcal{P}} \mid Y_{1:j-1}))$ bits of information because $Y_j$ possesses more information than is necessary to encode $B_{\mathcal{P}}$. This waste can stem from bad hyperparameter selection (i.e., the partitions in $\mathfrak{P}$ are low entropy) or reduced uncertainty about $X$ due to previous approximate MECs. While the latter is typically desirable (as it indicates we've achieved low conditional entropy), the former can negatively impact performance [Schroeder de Witt et al., 2023].

To address this, we introduce a technique that we call merging. At each iteration $j$, after performing a coupling, merging groups the possible realizations of $Y_j$ by the posterior update over $B_{\mathcal{P}}$ they induce. Instead of sampling a realization of $Y_j$, merging samples one of these groups. If the sampled group contains multiple elements, merging performs an additional coupling between the posterior over that group and the new maximum-entropy partition. This process repeats until the sampled group consists of a single element, at which point iteration $j + 1$ begins.

**An Example of Merging** To illustrate this process, consider a case in which

$$\nu(Y_j \mid Y_{1:j-1}) = [1/4, 1/4, 1/2],$$

$$\gamma(B_{\mathcal{P}} \mid Y_{1:j-1}) = [1/2, 1/2].$$

Then the following is a minimum-entropy coupling:

|       | $y_1$ | $y_2$ | $y_3$ |
|-------|-------|-------|-------|
| $b_1$ | 1/4   | 1/4   | 0     |
| $b_2$ | 0     | 0     | 1/2   |

In this coupling, the posterior over $B_{\mathcal{P}}$ remains the same whether $Y_j$ is realized as $y_1$ or $y_2$ (specifically, the probability of $b_1$ is one), indicating wasted entropy over $y_1, y_2$. Merging post-processes such couplings to yield:

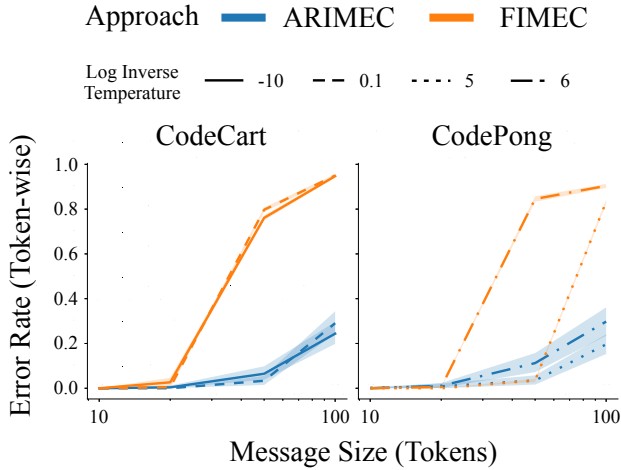

Figure 5: Results for the Markov coding games CodeCart and CodePong using MaxEntRL policies with different temperatures with 95% bootstrap confidence intervals drawn from 100 games.

|       | $\{y_1, y_2\}$ | $\{y_3\}$ |
|-------|----------------|-----------|
| $b_1$ | 1/2            | 0         |
| $b_2$ | 0              | 1/2       |

Under merging, if the group $\{y_1, y_2\}$ is sampled, the subsequent coupling is performed between the new maximum-entropy partition and

$$\nu(Y_j \mid Y_{1:j-1}, Y_j \in \{y_1, y_2\}).$$

If the sampled group has more than two elements, this process may repeat multiple times before proceeding to iteration $j + 1$.

## 6 EXPERIMENTS

To demonstrate the effectiveness of ARIMEC and merging, we perform experiments in two settings: Markov coding games [Sokota et al., 2022] and steganography [Cachin, 1998].

### 6.1 MARKOV CODING GAMES

In a Markov coding game (MCG) [Sokota et al., 2022], the goal is to communicate messages via the trajectories of a Markov decision process (MDP), while simultaneously achieving a high expected return in the MDP. Messages are sampled independently from a distribution known to both the player sending them and the player receiving them. For a more complete description, see Appendix D.2.

Sokota et al. [2022] give a principled approach to this setting called MEME that works in two steps. First, MEME trains a maximum-entropy reinforcement learning (MaxEntRL)

[Ziebart et al., 2008] policy for the MDP. (The intuition is that this policy balances between performing well in the MDP and having high bandwidth through which information can be communicated.) Second, MEME computes (or approximates) a minimum-entropy coupling between the distribution over messages and, roughly speaking, the distribution over trajectories induced by the MaxEntRL policy.[4] MEME guarantees that the expected return in the MCG is the same as in the MDP; furthermore, at each time step, MEME greedily maximizes the amount of information encoded into the trajectory. For a more complete description, see Appendix D.3.

Because the second step of MEME requires computing or approximating a MEC, prior to this work, it was only applicable to MCGs whose message distributions had small or factorable supports. Thus our extension of IMEC to arbitrary distributions also serves as an extension of MEME to arbitrary MCGs.

To illustrate the benefits of MEME's extended applicability, we perform experiments in two MCGs based on Cartpole and Pong [Bellemare et al., 2013], which we call CodeCart and CodePong, that were previously beyond MEME's applicability. For these MCGs, the distribution over messages is dictated by GPT-2 [Radford et al., 2019] with top-50 sampling. For each game, we trained two policies with using different entropy bonus temperatures that each achieved perfect scores in 100 of 100 games. As a baseline, we compare against a naive version of MEME that assumes that the message was sampled from a uniform distribution over tokens and uses FIMEC. Note that this baseline sacrifices MEME's expected return guarantee.

We show the rate at which trajectories are decoded incorrectly for each variant of IMEC in these settings (Figure 5). While both FIMEC and ARIMEC maintain perfect expected return in the MDP, ARIMEC produces substantially more efficient encodings.

## 6.2 STEGANOGRAPHY

In steganography, the goal is to encode information (called plaintext) into innocuous-seeming content (called stegotext), such that an adversary would not realize that the innocuous-seeming content contains hidden information. We consider two kinds of steganography for our experiments.

**Information-Theoretic Steganography** The first is information-theoretic steganography [Cachin, 1998], which seeks formal security guarantees. Schroeder de Witt et al. [2023] proved that this problem can be reduced to minimum-entropy coupling distributions of ciphertext (random bit-

<hr>

[4]To be more precise about the latter distribution requires nuance since environment transitions cannot be correlated with the message. See Sokota et al. [2022] for details.

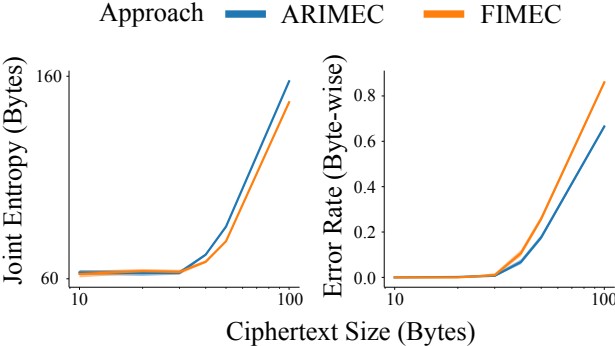

Figure 6: Results for information-theoretic steganography with 95% bootstrap confidence intervals drawn from 100 samples.

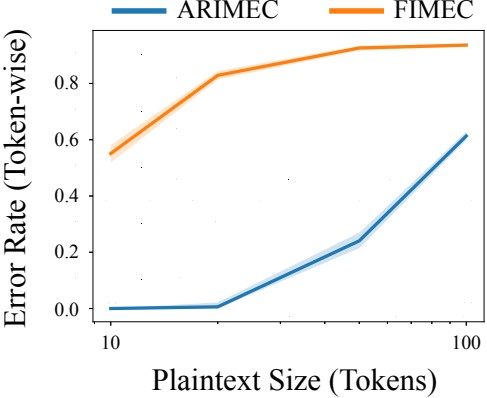

Figure 7: Results for unencrypted steganography with 95% bootstrap confidence intervals drawn from 100 samples.

strings generated using shared private keys) with distributions of covertext (innocuous content). For a more complete description, see Appendix D.4.

In this setting, Assumption 2.4 holds; thus, we would expect FIMEC to perform well relative to the ARIMEC. We show both the resulting joint entropy and the rate at which the ciphertext is decoded incorrectly in Figure 6, using 100 tokens sampled from GPT-2 as the covertext. This error rate can be written as $\mathbb{E}_{X \sim \mathcal{X}} \mathbb{E}_{Y \sim \gamma(Y|X)} I[X \neq \arg\max_x \gamma(x \mid Y)]$. Interestingly, while FIMEC produces lower joint entropy than ARIMEC, ARIMEC appears to produce a lower error rate. This could be because the ARIMEC focuses on maximizing the certainty of the bytes earlier in the string, while FIMEC focuses on reducing the uncertainty about the most uncertain bytes.

**Linguistic Steganography** The second setting is linguistic steganography, a broader concept than language-based information-theoretic steganography. Unlike the latter, linguistic steganography does not necessarily involve shared private keys. While not using private keys results in less

robust security guarantees, it offers two significant advantages. First, applicability is widened to settings in which a private key exchange is not possible. Second, potential information throughput is much higher, as the sender can use realistic priors about plaintext messages, which have significantly lower entropy compared to the uniform distribution of ciphertexts in information-theoretic steganography.

In the specific setting we consider, we aim to encode the output of one language model into that of another language model. The first language model acts as a prior on the plaintext messages that the sender may send, while the second language model serves as an approximate covertext distribution. In practice, one could achieve high throughput by first constructing a desired plaintext message and then translating it into a semantically equivalent message with a high likelihood under the prior.

For our experiments, the covertext distribution is generated by sampling 100 tokens from GPT-2 with the prompt "Here's an innocuous message:" and the plaintext message distribution is generated by GPT-2 with the prompt "Here's a secret message:". We compared ARIMEC with the correct prior against FIMEC under an assumed uniform prior over tokens. The results of this experiment are shown in Figure 7. Our findings indicate that ARIMEC substantially outperforms FIMEC in terms of information throughput, reflecting its ability to leverage the prior.

### 6.3 MERGING

To evaluate the performance of our merging technique, we consider a setting in which the objective is to transmit 10 bytes of ciphertext via GPT-2 stegotext. The results of this experiment, which we conducted using FIMEC, are depicted in Figure 8. The y-axis represents the joint entropy in bits, while the x-axis shows the dimension of the random vector $X$—i.e., $n$ in our notation.

As discussed in Section 5, poor choices of partition sets can negatively impact the performance of IMEC. In this case, FIMEC's performance significantly decreases (i.e., joint entropy increases) with the number of components, even though the entropy of $X$ is held constant. However, our merging technique substantially reduces IMEC's vulnerability to this issue, as desired.

### 7 CONCLUSION AND FUTURE WORK

In this work, we investigated the problem of computing low-entropy couplings for large-support distributions, making four main contributions. First, we unify existing algorithms under the formalism of partition sets. Second, using this unified perspective, we introduce ARIMEC—the first approach to computing low-entropy couplings for large-support distributions that can be applied to arbitrary distributions. Third,

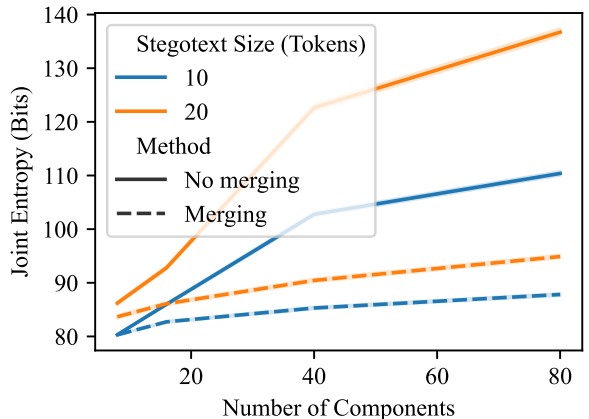

Figure 8: Comparing merging and not merging with 95% bootstrap confidence intervals drawn from 1000 samples.

we increase the robustness of IMEC algorithms to the choice of partition set by introducing a merging technique. Finally, we empirically show the utility of these innovations in MCG and steganography applications.

For future work, there are at least two application directions in which it would be interesting to push further with ARIMEC and merging. First is linguistic steganography. This direction is promising because ARIMEC can achieve high throughput rates, as we observed in Figure 7, and because of the recent proliferation of effective language models. Thus, there may be real-world settings in which it is applicable. Second, because ARIMEC is the first IMEC algorithm capable of handling arbitrary discrete distributions, it potentially opens the door to using large-support distributions for other minimum-entropy coupling applications in which the distributions may not be factorable, such as entropic causal inference, random number generation, functional representations, and dimensionality reduction.

### 8 ACKNOWLEDGEMENTS

This work was supported by ONR grant #N000142212121.

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

## A   INVERSE GENERATIVE PROCESSES

---

**Algorithm 5** Tabular IMEC: $X \mid Y = y$

---

**procedure** TIMEC($\mu, \nu, y$)
    $\gamma(X) \leftarrow \mu(X)$
    **for** $j = 1, \ldots, m$ **do**
        $\gamma(X, Y_j \mid y_{1:j-1}) \leftarrow \text{MEC}(\gamma(X \mid y_{1:j-1}), \nu(Y_j \mid y_{1:j-1}))$
    **end for**
    $X \sim \gamma(X \mid y)$
    return $X$
**end procedure**

---

---

**Algorithm 6** Factored IMEC: $X \mid Y = y$

---

**procedure** FIMEC($\mu, \nu, y$)
    $\gamma(X) \leftarrow \mu(X)$
    **for** $j = 1, \ldots, m$ **do**
        $i^* \leftarrow \text{argmax}_i \mathcal{H}(\gamma(X_i \mid y_{1:j-1}))$
        $\gamma(X_{i^*}, Y_j \mid y_{1:j-1}) \leftarrow \text{MEC}(\gamma(X_{i^*} \mid y_{1:j-1}), \nu(Y_j \mid y_{1:j-1}))$
        $\gamma(X, Y_j \mid y_{1:j-1}) \leftarrow \gamma(X_{i^*}, Y_j \mid y_{1:j-1}) \cdot \left( \prod_{i \neq i^*} \gamma(X_i \mid y_{1:j-1}) \right)$
    **end for**
    $X \sim \gamma(X \mid y)$
    return $X$
**end procedure**

---

---

**Algorithm 7** IMEC (Generic Form): $X \mid Y = y$

---

**procedure** IMEC($\mu, \nu, y, \mathfrak{P}$)
    $\gamma(X) \leftarrow \gamma(\mu)$
    **for** $j = 1, \ldots, m$ **do**
        $\mathcal{P} \leftarrow \arg\max_{\mathcal{P} \in \mathfrak{P}} \mathcal{H}(\gamma(B_{\mathcal{P}} \mid y_{1:j-1}))$
        $\gamma(B_{\mathcal{P}}, Y_j \mid y_{1:j-1}) \leftarrow \text{MEC}(\gamma(B_{\mathcal{P}} \mid y_{1:j-1}), \nu(Y_j \mid y_{1:j-1}))$
    **end for**
    $X \sim \gamma(X \mid y)$
    return $X$
**end procedure**

---

## B   THEORY

### B.1   RUNTIME COMPLEXITY

**Proposition 3.3** (IMEC Runtime). *Given a polynomial-time function for computing the maximum-entropy posterior partition, IMEC can be implemented in polynomial time in* $\max_j |\mathbb{Y}_j|, \max_i |\mathbb{X}_i|, m, n$.

*Proof.* Let $M = \max_j |\mathbb{Y}_j|$ and $N = \max_i |\mathbb{X}_i|$.

It suffices to show that each of the operations in the main loop requires only polynomial time, as the main loop runs $m$ times.

- By assumption, the maximum-entropy posterior partition requires only polynomial time.
- Performing an approximate minimum-entropy coupling on distributions of size $O(\max(M, N))$ requires only polynomial time.
- Marginalizing a joint distribution for variables of support $O(M), O(N)$ requires only polynomial time.

- Sampling from a distribution of support $O(M)$ requires only polynomial time.

$\square$

**Corollary B.1** (TIMEC Runtime). *TIMEC can be implemented in polynomial time in* $\max_j |\mathbb{Y}_j|, |\mathbb{X}|, m$.

*Proof.* Let $M = \max_j |\mathbb{Y}_j|$. Per Proposition 3.3, it suffices to show that maximum-entropy posterior partition computation is a polynomial time operation. Per Lemma B.3, the partition of singletons is always maximum entropy. Thus, since computing the posterior over $X$ is polynomial time in $M, |\mathbb{X}|$, the result follows. $\square$

**Corollary B.2** (FIMEC Runtime). *Let Assumption 2.4 hold. Then FIMEC can be implemented in polynomial time in* $\max_j |\mathbb{Y}_j|, \max_i |\mathbb{X}_i|, m, n$.

*Proof.* Let $M = \max_j |\mathbb{Y}_j|$ and $N = \max_i |\mathbb{X}_i|$. Per Proposition 3.3, it suffices to show that maximum-entropy posterior partition computation is a polynomial time operation. Since computing the posterior over each block—and the entropy of that posterior—is polynomial in $N, M$, and there are only $n$ blocks, the result follows. $\square$

## B.2 COUPLING

**Proposition 3.1** (Coupling). *IMEC induces a coupling of $\mu$ and $\nu$.*

*Proof.* We proceed by induction on $m$. For the base case, consider $m = 1$. Then for any $y \in \mathbb{Y}$

$$\sum_{x \in \mathbb{X}} \gamma(x, y) = \sum_{x \in \mathbb{X}} \mu(x) \gamma(y \mid x) \tag{3}$$

$$= \sum_{\mathbb{B} \in \mathcal{P}^{(1)}} \sum_{x \in \mathbb{B}} \mu(x) \gamma(y \mid \mathbb{B}) \tag{4}$$

$$= \sum_{\mathbb{B} \in \mathcal{P}^{(1)}} \gamma(y \mid \mathbb{B}) \sum_{x \in \mathbb{B}} \mu(x) \tag{5}$$

$$= \sum_{\mathbb{B} \in \mathcal{P}^{(1)}} \gamma(y \mid \mathbb{B}) \mu(\mathbb{B}) \tag{6}$$

$$= \sum_{\mathbb{B} \in \mathcal{P}^{(1)}} \gamma(y, \mathbb{B}) \tag{7}$$

$$= \nu(y), \tag{8}$$

where $\mathcal{P}^{(m)}$ denotes the partition selected at step $m$. Step (3) follows from chain rule; step (4) follows by construction; step (7) follows by chain rule; step (8) follows by the definition of a coupling. Now, assume the result holds up to $m = \bar{m}$, and consider $m = \bar{m} + 1$. Observe, for any $y \in \mathbb{Y}$

$$\sum_{x \in \mathbb{X}} \gamma(x, y) = \sum_{x \in \mathbb{X}} \mu(x) \gamma(y_{1:\bar{m}} \mid x) \gamma(y_{\bar{m}+1} \mid x, y_{1:\bar{m}}) \tag{9}$$

$$= \sum_{\mathbb{B} \in \mathcal{P}^{(\bar{m}+1)}} \sum_{x \in \mathbb{B}} \gamma(y_{1:\bar{m}}, x) \gamma(y_{\bar{m}+1} \mid \mathbb{B}, y_{1:\bar{m}}) \tag{10}$$

$$= \sum_{\mathbb{B} \in \mathcal{P}^{(\bar{m}+1)}} \gamma(y_{\bar{m}+1} \mid \mathbb{B}, y_{1:\bar{m}}) \sum_{x \in \mathbb{B}} \gamma(y_{1:\bar{m}}, x) \tag{11}$$

$$= \sum_{\mathbb{B} \in \mathcal{P}^{(\bar{m}+1)}} \gamma(y_{\bar{m}+1} \mid \mathbb{B}, y_{1:\bar{m}}) \gamma(y_{1:\bar{m}}, \mathbb{B}) \tag{12}$$

$$= \sum_{\mathbb{B} \in \mathcal{P}^{(\bar{m}+1)}} \gamma(y, \mathbb{B}, y_{1:\bar{m}}) \tag{13}$$

$$= \nu(y). \tag{14}$$

Step (9) follows from chain rule; step (10) follows by construction; step (13) follows by chain rule; step (14) follows by definition of a coupling. $\square$

## B.3 GREEDINESS

**Proposition 3.2** (Greediness). *If the partition of singletons is in $\mathfrak{P}$, IMEC approximately minimizes $\mathcal{H}(X, Y_{1:j})$ subject to $\mu, \nu, \gamma(X, Y_{1:j-1})$ on the $j$th iteration, for each $j$.*

*Proof.* Consider that performing a coupling with the partition of singletons (or a partition that it is equivalent up to elements with zero probability) is equivalent to performing a partition with $\mathbb{X}$ itself. Then, invoking Lemma B.3, it suffices to show that the statement holds for $\mathbb{X}$.

To see this, first recall

$$\mathcal{H}(X, Y) = \mathcal{H}(Y \mid X) + \mathcal{H}(X)$$

Because the entropy of $X$ is fixed (as it is determined by its marginal $\mu$), minimum-entropy coupling is equivalent to minimum-conditional-entropy coupling. Then, note that, by chain rule, we have

$$\mathcal{H}(Y_{1:j} \mid X) = \sum_{k=1}^{j} \mathcal{H}(Y_k \mid X, Y_{1:k-1}) = \mathcal{H}(Y_j \mid X, Y_{1:j-1}) + \sum_{k=1}^{j-1} \mathcal{H}(Y_k \mid X, Y_{1:k-1}).$$

At iteration $j$, all terms below $j$ have already been determined. Thus, the rightmost summation term is fixed and minimizing $\mathcal{H}(X, Y_{j-1})$ is reduced to minimizing $\mathcal{H}(Y_j \mid X, Y_{1:j-1})$. By again invoking the equivalence between minimum-entropy coupling and minimum-conditional-entropy coupling, this is equivalent to minimizing $\mathcal{H}(X, Y_j \mid Y_{1:j-1})$, which is exactly what IMEC minimizes at iteration $j$. $\qquad\square$

## B.4 CONDITION SATISFACTION FOR SPECIAL CASES

**Lemma B.3.** *Let $\mathfrak{P}$ be the set of all partitions over $\mathbb{X}$. For any distribution over $\mathbb{X}$, any maximum-entropy partition is equivalent to the partition of singletons up to zero-probability elements.*

*Proof.* Consider a block $\mathbb{B}$ of some partition $\mathcal{P}$ of $\mathbb{X}$. The entropy that $\mathbb{B}$ contributes is

$$-\gamma(\mathbb{B}) \log \gamma(\mathbb{B}).$$

The first derivative of this function is

$$-\log \gamma(\mathbb{B}) - 1.$$

The second derivative is

$$-\frac{1}{\gamma(\mathbb{B})}.$$

Since the second derivative is always negative, the contribution of $\mathbb{B}$ to the total entropy is strictly concave. Thus, further subdividing $\mathbb{B}$ increases its contribution to the total entropy, up to elements with zero probability. $\qquad\square$

## B.5 POSTERIOR UPDATES

**Lemma B.4** (Posterior Updates). *Let $(\mathbb{V}, \mathbb{E})$ be the prefix tree for $\mathbb{X}$. Assume that the posterior over a partition is updated if and only if its corresponding node is touched and that nodes are touched by traversing edges of the tree (without jumps). Let $\mathcal{P}_v$ be a partition whose posterior was updated on iteration $j$. If $v, u$ are neighbors and $u$ was last visited on iteration $j' \leq j$, then*

$$\gamma(\mathbb{B}_{u \to v} \mid Y_{1:j}) = 1 - \gamma(\mathbb{B}_{v \to u} \mid Y_{1:j})$$

*and, for $\mathbb{B}' \in \mathcal{P}_u$ such that $\mathbb{B}' \neq \mathbb{B}_{u \to v}$,*

$$\gamma(\mathbb{B}' \mid Y_{1:j}) \propto \gamma(\mathbb{B} \mid Y_{1:j}).$$

*Proof.* First consider that $\mathbb{B}_{u\to v}, \mathbb{B}_{v\to u}$ are pairs of complementary events. Thus, their probabilities must sum to one by the complement rule.

Now, consider that, if $u$ was last visited on iteration $j'$, it follows that no element of $\mathbb{B}_{v\to u}$ can have been visited since iteration $j'$. (This follows because every path from $\mathbb{B}_{v\to u}$ to $v$ must touch $u$ by definition of a tree.) Therefore, every partition updated since $\mathcal{P}_u$ was last updated must correspond to a vertex in $\mathbb{V}_{u\to v}$. Partitions corresponding to vertices in $\mathbb{V}_{u\to v}$ can only influence the blocks of $\mathcal{P}_u$ via $\mathbb{B}_{v\to u}$. Thus, because $\mathbb{B}_{v\to u} = \cup_{\mathbb{B}\in\mathcal{P}_u\setminus\mathbb{B}_{u\to v}}\mathbb{B}$, direct evidence about $\mathbb{B}_{v\to u}$ changes the probability of each element of $\mathcal{P}_u \setminus \mathbb{B}_{u\to v}$ by the same factor. $\qquad\square$

**Proposition 4.1** (Posterior Updates). *Assume that the posterior over a partition is updated if and only if its corresponding node is touched and that nodes are touched by traversing edges of the tree (i.e., without jumps). Let $\mathcal{P}_v$ be a partition whose posterior was updated on iteration $j$. If $v, u$ are neighbors and $u$ was last visited on iteration $j' \leq j$, then the iteration $j$ posterior for any partition $\mathcal{P}_u$ can be computed in polynomial time in $\max_i |\mathbb{X}_i|, n$.*

*Proof.* Let $N = \max_i |\mathbb{X}_i|$. If $u$ is a neighbor of $v$, then, using Lemma B.4, the posterior over $\mathcal{P}_u$ can be computed in $O(\max_i |\mathbb{X}_i|)$ time. If $u$ is not a neighbor of $v$, then we can compute the posterior over $\mathcal{P}_u$ by iteratively applying Lemma B.4 along the path from $v$ to $u$. Because path length is upper bounded by $O(n)$, the total time is polynomial in $\max_i |\mathbb{X}_i|, n$. $\qquad\square$

## B.6 ENTROPY UPPER BOUND

**Lemma B.5** (Entropy Upper Bound). *Let $\mu$ be a probability distribution over $\kappa$ elements. Fix any element $\mu(x^*)$. Then for any $q$ such that $\frac{1}{\kappa} \leq q \leq \mu(x^*)$, we have*

$$\mathcal{H}(\mu) \leq \begin{cases} -q\log q - (1-q)\log\frac{(1-q)}{\kappa-1} & q \in [1/\kappa, 1) \\ 0 & q = 1. \end{cases}$$

*Proof.* First note that if $\mu(x^*) = 1$ then $\mathcal{H}(\mu) = 0$ and the upper bound holds trivially.

Next, consider the case in which $\mu(x^*) < 1$. We will show that this upper bound holds in the case when $q = \mu(x^*)$. We first observe that the entropy is given by

$$\mathcal{H}(\mu) = -\sum_x \mu(x)\log\mu(x) = -\mu(x^*)\log\mu(x^*) - \sum_{x\neq x^*}\mu(x)\log\mu(x)$$

Now, we can consider another probability distribution $\mu'$ over $n-1$ values (everything except $x^*$), which is given by $\mu'(x) = \frac{\mu(x)}{1-\mu(x^*)}, \forall x \neq x^*$. Since entropy is maximized by a uniform distribution, we have that $\mathcal{H}(\mu') \leq -\log(\frac{1}{n-1})$.
We observe that

$$\begin{aligned} \mathcal{H}(\mu') &= -\sum_{x\neq x^*}\mu'(x)\log\mu'(x) \\ &= -\frac{1}{(1-\mu(x^*))}\sum_{x\neq x^*}\mu(x)\log\mu'(x) \\ &= -\frac{1}{(1-\mu(x^*))}\sum_{x\neq x^*}\mu(x)\Big(\log\mu(x) - \log(1-\mu(x^*))\Big) \\ &= -\frac{1}{(1-\mu(x^*))}\left[\sum_{x\neq x^*}\Big(\mu(x)\log\mu(x)\Big) + \log(1-\mu(x^*))\right] \end{aligned}$$

Then, plugging this into the inequality for $\mathcal{H}(\mu')$ gives us that

$$-\sum_{x\neq x^*}\mu(x)\log\mu(x) \leq (1-\mu(x^*))\left(-\log\left(\frac{1}{n-1}\right) - \log(1-\mu(x^*))\right)$$

$$= -(1-\mu(x^*))\log\left(\frac{1-\mu(x^*)}{n-1}\right)$$

Thus, this gives us that

$$\mathcal{H}(\mu) \le -\mu(x^*)\log\mu(x^*) - (1-\mu(x^*))\log\left(\frac{1-\mu(x^*)}{n-1}\right)$$

as desired.

Next, we will show that this upper bound decreases in $q$. We can consider taking the partial derivative with the upper bound with respect to $q$, which gives us that

$$D_q\left(-q\log q - (1-q)\log\frac{(1-q)}{n-1}\right) = -\log q - 1 + \log\frac{(1-q)}{n-1} + 1 = -\log q + \log\frac{(1-q)}{n-1}.$$

Setting this equal to zero gives us that

$$\log q - \log\frac{1-q}{n-1} = 0$$

$$\implies q = \frac{1}{n}.$$

Next, we observe that the second derivative of the upper bound with respect to $q$ is given by

$$D_q D_q\left(-q\log q - (1-q)\log\frac{(1-q)}{n-1}\right) = D_q\left(-\log q + \log\frac{(1-q)}{n-1}\right) = \frac{1}{q(q-1)}.$$

Thus, this is negative for all values of $0 < q < 1$, which gives us that the upper bound is decreasing in $q$ on the interval $[\frac{1}{n}, 1)$. Therefore, since it holds for $q = \mu(x^*)$, it must hold for $q \in [1/n, \mu(x^*)]$. $\qquad\square$

**Proposition 4.2** (Maximum-Entropy Partition). *Let*

$$\kappa \ge \max_{\mathcal{P}\in\mathfrak{P}}|\{\mathbb{B}\in\mathcal{P}\mid\mu(\mathbb{B}) > 0\}|$$

*be an upper bound on the number of blocks with positive probability. Define*

$$\mathcal{U}: q \mapsto -q\log q - (1-q)\log\frac{1-q}{\kappa-1}.$$

*For any neighbor $u$ of $v$, if*

$$\gamma(\mathbb{B}_{v\to u}\mid y_{1:j}) < 1 - 1/\kappa,$$

*then, for all $u' \in \mathbb{V}_{v\to u}$,*

$$\mathcal{H}(\gamma(B_{\mathcal{P}_{u'}}\mid y_{1:j})) \le \mathcal{U}(\gamma(\mathbb{B}_{u\to v}\mid y_{1:j})).$$

*Proof.* Observe

$$\gamma(\mathbb{B}_{v\to u}\mid Y_{1:j}) < 1 - 1/\kappa$$
$$\iff -\gamma(\mathbb{B}_{v\to u}\mid Y_{1:j}) > -1 + 1/\kappa$$
$$\iff 1 - \gamma(\mathbb{B}_{v\to u}\mid Y_{1:j}) > 1/\kappa$$
$$\iff \gamma(\mathbb{B}_{u\to v}\mid Y_{1:j}) > 1/\kappa.$$

Fix any $u' \in \mathbb{V}_{v\to u}$. Then, $\mathbb{B}_{u\to v} \subset \mathbb{B}_{u'\to v}$. Therefore, we have $\gamma(\mathbb{B}_{u'\to v}\mid Y_{1:j}) \ge \gamma(\mathbb{B}_{u\to v}\mid Y_{1:j})$. The bound follows from applying Lemma B.5. $\qquad\square$

## C  VISUALIZATIONS

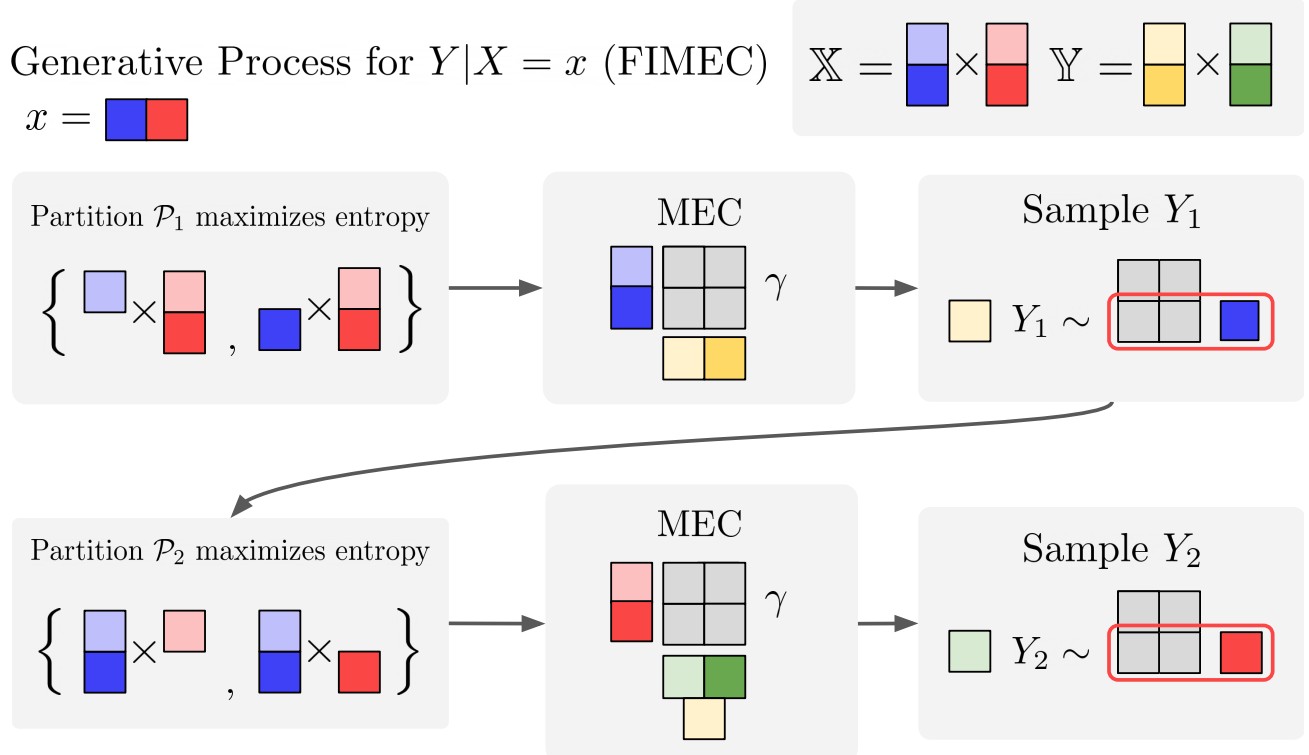

Figure 9: Visualization of two iterations of FIMEC.

For comparison to ARIMEC, Figure 9 shows two iterations of FIMEC.

## D  EXPERIMENTS

### D.1  MAXIMUM-ENTROPY PARTITION SEARCH

In Proposition 3.3, we demonstrated that instances of IMEC are efficient if and only if the maximum-entropy posterior partition can be computed efficiently. For ARIMEC, we established in Proposition 4.1 that the posterior of individual nodes can be computed efficiently. However, we did not prove that Algorithm 4 searches through only a polynomial number of nodes, raising concerns about the practical efficiency of ARIMEC. Fortunately, our empirical observations indicate that the search procedure is highly effective. To illustrate this, in Figure 10, we show the number of nodes the search procedure required, on average, to compute the maximum-entropy posterior partition as a function of the number of nodes in the prefix tree for two distributions: GPT-2 and random bytestrings. We find that, even as the prefix tree grows very large, average the number of nodes touched per iteration remains manageable.

### D.2  MARKOV CODING GAMES

Sokota et al. [2022] specify Markov coding games as the following setting:

> An MCG is a tuple $\langle(\mathcal{S}, \mathcal{A}, \mathcal{T}, \mathcal{R}), \mathcal{M}, \mu, \zeta\rangle$, where $(\mathcal{S}, \mathcal{A}, \mathcal{T}, \mathcal{R})$ is a Markov decision process, $\mathcal{M}$ is a set of messages, $\mu$ is a distribution over $\mathcal{M}$ (i.e., the prior over messages), and $\zeta$ is a non-negative real number we call the message priority. **An MCG proceeds in the following steps:**
>
> 1. First, a message $M \sim \mu$ is sampled from the prior over messages and revealed to the sender.

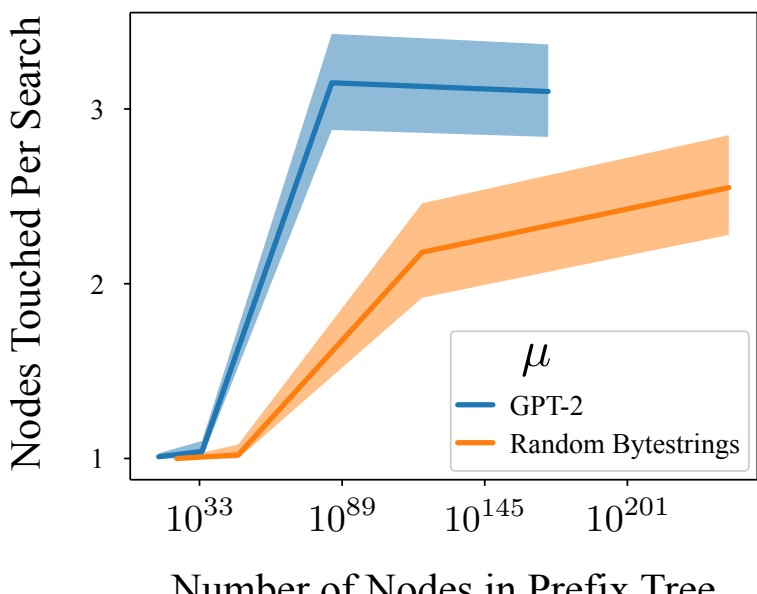

Figure 10: Results for number of nodes touched with 95% bootstrap confidence intervals drawn from 100 samples.

2. Second, the sender uses a message conditional policy $\pi_{|M}$, which takes states $s \in \mathcal{S}$ and messages $m \in \mathcal{M}$ as input and outputs distributions over MDP actions $\Delta(\mathcal{A})$, to generate a trajectory $Z \sim (\mathcal{T}, \pi_{|M})$ from the MDP.
3. Third, the sender's terminal MDP trajectory $Z$ is given to the receiver as an observation.
4. Fourth, the receiver uses a terminal MDP trajectory conditional policy $\pi_{|Z}$, which takes terminal trajectories $z \in \mathcal{Z}$ as input and outputs distributions over messages $\Delta(\mathcal{M})$, to estimate the message $\hat{M} \sim \pi_{|Z}(Z)$.

The objective of the agents is to maximize the expected weighted sum of the return and the accuracy of the receiver's estimate $\mathbb{E}\left[\mathcal{R}(Z) + \zeta \mathbb{I}[M = \hat{M}] \mid \pi_{|M}, \pi_{|Z}\right]$. Optionally, in cases in which a reasonable distance function is available, we allow for the objective to be modified to minimizing the distance between the message and the guess $d(M, \hat{M})$, rather than maximizing the probability that the guess is correct.

### D.3 MEME

Sokota et al. [2022] specify MEME as follows:

**Step One: Maximum Entropy Reinforcement Learning** In the first step, MEME uses MaxEnt RL to construct an MDP policy $\pi$. This policy is an MDP policy, not an MCG policy, and therefore does not depend on the message. Note that this policy depends on the choice of temperature $\alpha$ used for the MaxEnt RL algorithm.

**Step Two: Minimum Entropy Coupling** In the second step, at execution time, MEME constructs a message-conditional policy online using MECs. Say that, up to time $t$, the sender is in state $s^t$, history $h^t$ and has played according to the state and message conditional policy $\pi_{|M}^{:t}$ so far. Let

$$b^t = \mathcal{P}(M \mid h^t, \pi_{|M}^{:t})$$

be the posterior over the message, conditioned on the history and the historical policy. MEME performs a MEC between the posterior over the message $b^t$ and distribution over actions $\pi(s^t)$, as determined by the MDP policy. Let $\nu = \text{MEC}(b^t, \pi(s^t))$ denote joint distribution over messages and actions resulting from the coupling. Then MEME sets the sender to act according to the message conditional distribution

$$\pi_{|M}^t(s^t, m) = \nu(A^t \mid M = m)$$

of the coupling distribution $\nu = \text{MEC}(b^t, \pi(s^t))$.

Given the sender's MDP trajectory, MEME's receiver uses the sender's MDP policy and MEC procedure to reconstruct the sender's message conditional policy along the trajectory; thereafter, the receiver computes the posterior and guesses the maximum a posteriori (MAP) message.

## D.4 INFORMATION-THEORETIC STEGANOGRAPHY

Schroeder de Witt et al. [2023] summarize Cachin [1998]'s information-theoretic steganography setting as follows:

**Problem Setting** The objects involved in information-theoretic steganography can be divided into two classes: those which are externally specified and those which require algorithmic specification. Each class contains three objects. The externally specified objects include the distribution over plaintext messages $\mathcal{M}$, the distribution over covertext $\mathcal{C}$, and the random source generator.

- The distribution over plaintext messages $\mathcal{M}$ may be known by the adversary, but is not known by the sender or the receiver. However, the sender and receiver are aware of the domain $\mathbb{M}$ over which $\mathcal{M}$ ranges. The sampled plaintext message $M$ is explicitly known by the sender, but not to the receiver or the adversary.
- The covertext distribution $\mathcal{C}$ is assumed to be known by the sender, the receiver, and the adversary.
- The random source generator provides the sender with a mechanism to take random samples from distributions. This random source is known to the sender but not to the receiver or adversary.

The objects requiring algorithmic specification, which are collectively referred to as a stegosystem, are the key generator, the encoder, and the decoder.

- The key generator produces a private key $K$ in the form of a binary string. This private key is shared between the sender and receiver over a secure channel prior to the start of the stegoprocess and can be used to coordinate encryption and decryption. The key generation process may be known to the adversary, but the realization of the key $K$ is not.
- The encoder takes a private key $K$, a plaintext message $M$, and a source of randomness $R$ as input and produces a stegotext $S$ in the space of covertexts $\mathbb{C}$.
- The decoder takes a private key $K$ and a stegotext $S$ as input and returns an estimated plaintext message $\hat{M}$.

They specify the following objectives and methodological outline for the setting:

**Definition D.1.** *[Cachin, 1998] Given covertext distribution $\mathcal{C}$ and plaintext message space $\mathbb{M}$, a stegosystem is $\epsilon$-secure against passive adversaries if the KL divergence between the distribution of covertext $\mathcal{C}$ and the distribution of stegotext $\mathcal{S}$ less than $\epsilon$; i.e., $KL(\mathcal{C}, \mathcal{S}) < \epsilon$. It is perfectly secure if the KL divergence is zero; i.e., $KL(\mathcal{C}, \mathcal{S}) = 0$.*

In other words, a steganographic system is perfectly secure if the distribution of stegotext $\mathcal{S}$ communicated by the sender is exactly the same as the distribution of covertext $\mathcal{C}$.

In addition to security, it is desirable for stegosystems to transmit information efficiently. Mutual information between messages and stegotexts is one way to quantify efficiency.

**Definition D.2.** *The mutual information $\mathcal{I}(M; S) = \mathcal{H}(M) - \mathcal{H}(M \mid S)$ between the message $M$ and stegotext $S$ is the expected amount of uncertainty in the message $M$ that is removed by conditioning on the stegotext $S$.*

**Methodological Outline** A common class of stegosystems uses two-step encoding and two-step decoding processes, as described below:

1. The sender uses the private key $K$ to injectively map the plaintext message $M$ into ciphertext $\mathbb{X} = \{0, 1\}^\ell$ in such a way that the induced distribution over ciphertext $\mathcal{X}$ is uniformly random, regardless of the distribution of $\mathcal{M}$.[5]

---

[5]For example, if $K$ is drawn from a uniform random distribution, $\text{bin}(M)$ denotes a deterministic binarization of $M$, and XOR represents the component-wise exclusive-or function, then $X = \text{XOR}(\text{bin}(M), K)$ is guaranteed to be distributed uniformly randomly, regardless of the distribution of messages $\mathcal{M}$.

2. The sender uses a (potentially stochastic) mapping $f\colon \mathbb{X} \rightsquigarrow \mathbb{C}$ to transform the ciphertext $X$ into stegotext $S$ (which exists in the space of covertexts $\mathbb{C}$).

3. The receiver estimates the ciphertext $\hat{X}$ from the stegotext $S$.

4. The receiver inverts the estimated ciphertext $\hat{X}$ to a plaintext message $\hat{M}$ with private key $K$.[6]

Given the definition below Schroeder de Witt et al. [2023] show the following guarantees:

**Definition D.3.** *We say that an encoding procedure $f\colon \mathbb{X} \rightsquigarrow \mathbb{C}$ is induced by a coupling if there exists $\gamma \in \Gamma(\mathcal{X}, \mathcal{C})$ such that for all $x \in \mathbb{X}, c \in \mathbb{C}, \mathcal{P}(f(x){=}c) = \gamma(C{=}c \mid X{=}x)$.*

**Theorem D.4.** *A steganographic encoding procedure is perfectly secure if and only if it is induced by a coupling.*

**Theorem D.5.** *Among perfectly secure encoding procedures, a procedure $f\colon \mathbb{X} \rightsquigarrow \mathbb{C}$ maximizes the mutual information $\mathcal{I}(M; S)$ if and only if $f$ is induced by a minimum entropy coupling.*

---

[6]For the example in footnote 5, the receiver can recover the binarized message $\mathrm{bin}(M)$ using the mapping $X \mapsto \mathrm{XOR}(X, K)$ and invert the binarization to recover the plaintext $M$.