# OpenReview forum: "Computing Low-Entropy Couplings for Large-Support Distributions"
_auai.org/UAI/2024/Conference — UAI 2024 poster_

### Official Review · Reviewer_f2dV · 2024-03-05

**Q2-1 Originality-Novelty:** 3
**Q2-2 Correctness-Technical Quality:** 3
**Q2-5 Clarity Of Writing:** 4

**Q1 Summary And Contributions:**

The authors study couplings of minimum entropy and give some interesting results.

**Q2-3 Extent To Which Claims Are Supported By Evidence:**

4: Excellent: all claims are supported by very convincing evidence (in the form of comprehensive experimental evaluation, rigorous mathematical proofs, detailed (pseudo-)code, precise references, well-motivated and realistic assumptions) and the authors deliver what they promise.

**Q2-4 Reproducibility:**

4: Excellent: key resources (e.g. proofs, code, data) are available and key details (e.g. proof sketches, experimental setup) are comprehensively described for competent researchers to confidently and easily reproduce the main results.

**Q3 Main Strengths:**

The unification of IMEC algorithms under a single formalism is a very important contribution.

**Q4 Main Weakness:**

None.

**Q5 Detailed Comments To The Authors:**

Page 1:
How is steganography related to cryptography? :)

Page 2:
Can you please expand the discussion of Subsection 2.2?

Page 3:
In your algorithms, can you please clearly state their Input and Output?

Item 2:
How is this sampling done?

Page 4:
I do not understand Footnote 3 :)

Page 5:
Definition 4.4 could be more clear.

Page 6:
Please some more details about the implementation in the main body.

Page 7:
Please elaborate on the caption of Figure 3.

Page 8:
I like your Conclusions section :)

Can you please add some more discussion about future work/next steps?

**Q9 Complying With Reviewing Instructions:**

Yes

---

> ### Author Rebuttal · Authors · 2024-04-08
>
> > Page 1: How is steganography related to cryptography? :)
>
> Informally, in cryptography, we're ok with a third party seeing that we're sending encrypted messages as long as that third party is unable to decrypt those encrypted messages. In contrast, in steganography, we want to hide the fact that we're sending any sensitive information in the first place.
>
> > Page 2: Can you please expand the discussion of Subsection 2.2?
>
> Yes -- are there specific aspects it would be most helpful to expand? We are happy to provide more details in the Appendix.
>
> > Page 3: In your algorithms, can you please clearly state their Input and Output?
>
> As stated in the pseudocode of the algorithms in the main body, $\mu$, $\nu$ and $x$ are given as input and $Y$ is given as output. As discussed in Definition 2.1, $\mu$ and $\nu$ are the marginal probability distributions over which we are coupling, $x$ is a member of the set over which $\mu$ ranges, and $y$ is a member of the set over which $\nu$ ranges.
>
> > Item 2: How is this sampling done?
>
> In our code, we use the function `numpy.random.choice`. Other choices would also work -- we're just sampling from a discrete distribution with known probabilities.
>
> > Page 4: I do not understand Footnote 3 :)
>
>
> Footnote 3 discusses why coupling with the maximum-entropy partition leads to a large potential reduction in joint entropy. (We are trying to reduce the entropy as much as possible in order to get as close as possible to minimum entropy.) Specifically, Footnote 3 shows that we are maximizing an upper bound on the entropy reduction. Note that this is a heuristic approach in the sense that, ideally, we would want to maximize a lower bound on entropy reduction.
>
> > Page 5: Definition 4.4 could be more clear.
>
> Are there particlar aspects the reviewer found unclear? We are happy to add clarification.
>
> > Page 6: Please some more details about the implementation in the main body.
>
> We provide pseudocode for our general iterative minimum-entropy coupling approach, as well as the pseudocode for computing the maximum entropy partition. Does the reviewer have particular ideas on what other details might be helpful to include?
>
> > Page 7: Please elaborate on the caption of Figure 3.
>
> Figure 3 provides our experimental results on two Markov coding game tasks (see Subsection 5.1 and Appendix D for a description). We compare using our method (ARIMEC) with an existing iterative MEC approach (FIMEC) in an algorithm that encodes messages in the trajectory of a Markov decision process via computing minimum-entropy couplings.
>
> We observe that ARIMEC achieves a much lower token-wise error rate in terms of communicating this message, when compared to FIMEC, especially in cases with much larger message sizes. This happens because ARIMEC is able to leverage the message prior, whereas FIMEC is not.
>
> > Can you please add some more discussion about future work/next steps?
>
> We provide a discussion about potential future work, including on unencrypted steganography and extending other use cases of minimum-entropy coupling to large-support distributions, in the second paragraph in our Discussion (Section 6). Is the reviewer most interested in an expanded discussion on one of these in particular?

---

### Official Review · Reviewer_AhV5 · 2024-03-22

**Q2-1 Originality-Novelty:** 3
**Q2-2 Correctness-Technical Quality:** 3
**Q2-5 Clarity Of Writing:** 2

**Q1 Summary And Contributions:**

This paper studies the minimum-entropy coupling problem and unifies the existing iterative minimum-entropy coupling (IMEC) approach to derive a new IMEC method that can be applied to distributions with large support.

**Q2-3 Extent To Which Claims Are Supported By Evidence:**

3: Good: the main claims are supported by convincing evidence (in the form of adequate experimental evaluation, proofs, (pseudo-)code, references, assumptions).

**Q2-4 Reproducibility:**

2: Fair: key resources (e.g. proofs, code, data) are unavailable but key details (e.g. proof sketches, experimental setup) are sufficiently well-described for an expert to confidently reproduce the main results.

**Q3 Main Strengths:**

The paper provides a nice overview of the existing method.

The proposed method is derived rigorously.

Various experiments are performed to demonstrate the effectiveness of the proposed algorithm.

**Q4 Main Weakness:**

There could be some emphasis on comparing the proposed method with existing ones. For example, highlight the inapplicability of existing methods under the unified IMEC form.

**Q5 Detailed Comments To The Authors:**

To what extent the proposed method is applicable where other existing IMEC approaches are not?

How does the runtime of ARIMEC implemented by Algorithm 3 compare with the existing approaches?

**Q9 Complying With Reviewing Instructions:**

Yes

---

> ### Author Rebuttal · Authors · 2024-04-08
>
> > There could be some emphasis on comparing the proposed method with existing ones. For example, highlight the inapplicability of existing methods under the unified IMEC form. To what extent the proposed method is applicable where other existing IMEC approaches are not?
>
> TIMEC and FIMEC are only applicable in settings where one of the following is true:
> 1. One of the distributions being coupled has a small support.
> 2. One of the distributions being coupled is factorable into component distributions, each having a small support.
>
> In contrast, ARIMEC is applicable to any discrete distribution. In the current version of the submission, we discuss the contrast in applicability in:
> - The abstract: "We derive a new IMEC instance from this formalism, which we call ARIMEC, that, unlike existing IMEC algorithms, can be applied in practice to arbitrary discrete distributions."
> - Section 1: "As a result, at the time of writing, there exist no techniques for producing low-entropy couplings of general large-support distributions ... Leveraging this formalism, we derive the first algorithm for computing low-entropy couplings for arbitrary large-support distributions, which we call autoregressive IMEC (ARIMEC)."
> - Section 4: "ARIMEC improves upon the applicability of FIMEC by eliminating the factorability assumption, which does not hold in general."
> - Section 5: "Because the second step of MEME requires computing or approximating a MEC, prior to this work, it was only applicable to MCGs whose message distributions had small or factorable supports."
> - Section 6: "ARIMEC—the first approach to computing low-entropy couplings for large-support distributions that can be applied to arbitrary distributions."
>
> Please let us know whether it would be helpful to add further emphasis on this point.
>
>
> > How does the runtime of ARIMEC implemented by Algorithm 3 compare with the existing approaches?
>
> It's a bit tricky to give a good answer for this because they're applicable to different problem settings. In settings where TIMEC is applicable, ARIMEC and TIMEC are equivalent (and so would have the same runtime). We can't make such precise statements when comparing with FIMEC (there isn't an analogous equivalencey). In our experiments, we found that ARIMEC was often at least a factor of 2 slower on factorable distributions (which is where FIMEC is applicable). However, our implementation of ARIMEC is general, meaning that it doesn't leverage the assumption that one of the distributions is factorable. If we were to use an implementation that did leverage this assumption, we expect that much of this gap would be closed.

---

### Official Review · Reviewer_476S · 2024-03-22

**Q2-1 Originality-Novelty:** 3
**Q2-2 Correctness-Technical Quality:** 3
**Q2-5 Clarity Of Writing:** 3

**Q1 Summary And Contributions:**

The paper studies couplings, that is bivariate joint distributions for given marginals, for discrete distributions with large support.
Here the focus is to find low-entropy couplings, i.e.~couplings with small (or minimal) Shannon entropy.
The paper claims contributions in the area of algorithmic improvements.
In particular, it unifies and generalizes previous methods and introduces a new approach for arbitrary marginal distributions with large support.
The performance of the methods is studied empirically.

**Q2-3 Extent To Which Claims Are Supported By Evidence:**

3: Good: the main claims are supported by convincing evidence (in the form of adequate experimental evaluation, proofs, (pseudo-)code, references, assumptions).

**Q2-4 Reproducibility:**

3: Good: key resources (e.g. proofs, code, data) are available and key details (e.g. proofs, experimental setup) are sufficiently well-described for competent researchers to confidently reproduce the main results.

**Q3 Main Strengths:**

The paper generalizes and formalizes previous approaches and extends this to the large-support setting.

**Q4 Main Weakness:**

For someone who is not familiar with this line of research the background section is too brief.
In particular it would be helpful if sections 2.3 and 2.4 would be expanded.
In general, the paper seems to require familiarity with Sokota et al. (2022) and Schroeder de Witt et al. (2023).

**Q5 Detailed Comments To The Authors:**

1. Consider explaining in more detail the background in sections 2.3 and 2.4.
2. factorable = pairwise independent?
3. Could you expand on the link to autoregressive models in Section 4?

**Q9 Complying With Reviewing Instructions:**

Yes

---

> ### Author Rebuttal · Authors · 2024-04-08
>
> > Consider explaining in more detail the background in sections 2.3 and 2.4.
>
> Thanks for this feedback! In an earlier draft, we included a more detailed Section 2.3 and 2.4 but decided it was taking too long to get to the contribution of the submission. Are there particular aspects of 2.3 and 2.4 that the reviewer thinks would be most economical to expand on?
>
> > factorable = pairwise independent?
>
> By factorable, we mean $P(X_1, \dots, X_n) = P(X_1) \cdot \dots \cdot P(X_n)$. We will make this more explicit in the revised version of the submission.
>
> > Could you expand on the link to autoregressive models in Section 4?
>
> Yes, by autoregressive model, we mean a model $p_{\theta}$ of some random vector $X=(X_1, \dots, X_n)$ where, for any $i$ and any $x_1, \dots, x_{i-1}$, $p_{\theta}$ defines a mapping from $(x_1, \dots, x_{i-1})$ to $\Delta(\mathbb{X}_i)$. This is the form that, say, transformer models often take. We will make this more explicit in the revised version of the submission.

---

### Meta-Review · Area_Chair_o7v8 · 2024-04-17

The authors propose a unification of recent work on (approximately) minimum entropy couplings.  The reviewers tended towards accept, but with relatively low confidence.  I believe that the work does present important theoretical, if not, practical contributions that adds sufficient novelty on top of existing work (despite the low confidence reviews).  I would encourage the authors to incorporate some of the feedback (especially in terms of presentation of related work) into consideration when revising the draft.